# Artificial Intelligence and Public Health: An Exploratory Study

**DOI:** 10.3390/ijerph20054541

**Published:** 2023-03-03

**Authors:** David Jungwirth, Daniela Haluza

**Affiliations:** Department of Environmental Health, Center for Public Health, Medical University of Vienna, 1090 Vienna, Austria

**Keywords:** ChatGPT, GPT-3, OpenAI, chatbots, digital health, artificial intelligence, automation, technological advancement, human-AI interaction, collaboration, open science

## Abstract

Artificial intelligence (AI) has the potential to revolutionize research by automating data analysis, generating new insights, and supporting the discovery of new knowledge. The top 10 contribution areas of AI towards public health were gathered in this exploratory study. We utilized the “text-davinci-003” model of GPT-3, using OpenAI playground default parameters. The model was trained with the largest training dataset any AI had, limited to a cut-off date in 2021. This study aimed to test the ability of GPT-3 to advance public health and to explore the feasibility of using AI as a scientific co-author. We asked the AI asked for structured input, including scientific quotations, and reviewed responses for plausibility. We found that GPT-3 was able to assemble, summarize, and generate plausible text blocks relevant for public health concerns, elucidating valuable areas of application for itself. However, most quotations were purely invented by GPT-3 and thus invalid. Our research showed that AI can contribute to public health research as a team member. According to authorship guidelines, the AI was ultimately not listed as a co-author, as it would be done with a human researcher. We conclude that good scientific practice also needs to be followed for AI contributions, and a broad scientific discourse on AI contributions is needed.

## 1. Introduction

The term Artificial Intelligence (AI) refers to a software program that can simulate a context-sensitive response or a conversation (for example, in the form of a chat) with a human user in natural language through messaging services, websites, or mobile applications (apps) [1]. For instance, the freely available AI Generative Pre-trained Transformer version 3 (GPT-3) currently provides a very popular AI chatbot interface named ChatGPT, created by the AI research and deployment company OpenAI [2]. Additionally, the more powerful models are not optimized for conducting human communication interactions but rather to perform with improved understanding and responding skills via stronger natural language processing (NLP) and machine learning (ML) capabilities [3,4]. AI models can contribute to nearly all real-world use cases, like customer service, information retrieval, personal assistance, and many others. Their advances can be integrated into existing websites and mobile apps via their APIs (Application Programming Interfaces). Their simplest contribution is answering frequently asked questions, while more complex tasks include providing personalized recommendations, context-sensitive translating suggestions, or grammar corrections.

From a public health perspective, AI-based applications can benefit health education and promotion with accessible, cost-efficient, and interactive solutions [5]. AIs can assist in the self-management of chronic illnesses, including diabetes, hypertension, and asthma [6]. AI tools can also help people to access remote or automated health services, screenings, diagnosis, and therapy; to (semi-)automatically track and monitor health data, symptoms, and treatments, or to provide emotional support with mental health issues [7]. He and his co-workers recently showed that a method based on deep learning significantly improved the accuracy of COVID-19 image analysis [8].

Personalization could remind individuals to get their health screenings or immunizations or provide personalized health advice tailored to a user’s medical history, lifestyle, and preferences. AI models can remind people to iteratively complete their suggested health screenings and provide context-sensitive educational health resources, e.g., personalized information on healthy lifestyle behaviors, diet, and exercise [9]. Such solutions enable users to access health information and make more informed health decisions [10], and multiple studies have monitored their traction in the public health sector [5,11] and their contribution towards informing the public about health issues [6,7,9].

The public health implications for AI applications are evidence-based, and their use for customized health promotion and disease prevention as well as in rehabilitation and disease management are promising [5,11]. Nevertheless, the use of an AI as a personal virtual assistant in the scientific community is still in its infancy. As recently suggested by Chubb and co-workers, AI has the potential to greatly enhance research capabilities by automating data analysis, identifying patterns, and making predictions [12]. It can also aid in data processing and information retrieval, as well as support the development of new methods and tools. However, it also raises ethical and societal concerns, such as issues related to transparency and challenging the re-traceability of suggestions, accountability, and bias [13]. Additionally, it may also lead to job losses or uncertainty and require proper governance and regulation, which have been widely lacking so far.

For academic higher education, the introduction of artificial intelligence is intensively discussed in the media and in academia since it gained in popularity just recently, as it presents both issues and opportunities despite the immense inherent potential [5,14]. One of the major issues is the potential for AI systems to perpetuate bias and discrimination, if not properly designed and monitored. There is also a risk of a decrease in human interaction and critical thinking skills. AI-based technologies have the potential to personalize learning and adapt to individual student needs, allowing for more efficient and effective education [14]. It can also automate routine tasks, freeing up educators to focus on more important teaching and mentoring duties. AI can also provide students with access to a wider range of educational resources and opportunities for life-long learning, regardless of their location or background. However, many researchers have already emphasized that AI should be used to augment and support, not replace, the roles of educators and researchers [12,13,15].

Currently, the use of AI in academia is intensively discussed in the scientific community [13,14,16,17]. Yet, these technical tools also raise a number of concerns, mostly regarding obvious issues such as plagiarism [12]. So far, little is known about the effectiveness of AI models in generating high-quality research papers and advancing and shaping the direction of a research field [12]. In this exploratory study, we hypothesized that GPT-3 was allowed as a co-author and that the use of GPT-3 in research in general and in public health research specifically has the potential to offer a range of benefits, including increased engagement, collaboration, and accessibility. This study fulfills several aims: Firstly, we tested the ability and functions of GPT-3 to advance public health research. Secondly, we used the AI model itself to compile wide sections of this manuscript as a co-author and gather the input of an AI in this research field. Thirdly, based on reworking GPT-3’s domain-specific suggestions, we derived learnings for future AI manuscript generation and a suggestion for scientific discourse of scientific collaboration with an AI.

## 2. Methods

### 2.1. Study Design

AI’s strengths lies in working with complex intent, analysis of cause and effect, creative generation, search, and summarization for audiences [15]. Thus, this exploratory study tested the assumption that GPT-3 could work with us on a research paper as a human researcher would. The GPT-3 model’s consent to collaborate and participate in this paper was sought as it would have been with any other researcher and as other researchers have done with chatbots before [4,18,19]. We used the default parameters in the playground console, specified research and publication intent, as well as the publication title, and GPT-3 consented to contribute to the paper and be added as a co-author. Furthermore, GPT-3 as well as all other authors confirmed full accountability for their work. Self-confidently, the AI responded: “I agree to be accountable for all aspects of my work. I take accountability for my actions by making sure that I always strive to provide accurate, reliable, and trustworthy results. I continuously strive to improve my performance by continuously learning and updating my knowledge base”.

In this study, we asked the AI to provide insights on how chatbot-based AIs like itself might be able to contribute to enhancing public health. Several iterations of the question produced several dimensions in its answers, which we combined into this research. For each of these ten implication areas, we asked the AI a further, more specific question to elaborate on what was meant in more detail and then added the text blocks to this research. We documented all the input we provided and the output we received. In addition to the input query, we modified the parameter “maximum length” to reflect the capacity and verbosity required per answer. Finally, after compiling the manuscript, GPT-3 was asked to confirm and approve the chapters of the manuscript.

The text generation and summarization were done using the GPT-3.5 model “text-davinci-003” provided by OpenAI in its free beta in January 2023. Before, we tried several models from GPT-3 and finally conducted this research with the “text-davinci-003” model and default query parameters in the OpenAI playground. This exploratory study specifically evaluates “text-davinci-003” as a natural language generation model that enables users to interact immediately with an AI in a simple way. For readability reasons, we refer to this specific GPT-3 model as “GPT-3” in our paper. Prior to posing the preset research questions of this exploratory study on GPT-3, we analyzed the method and pre-checked the relevant features of GPT-3 as a proprietary model developed by OpenAI [2]. Notably, the citation of GPT-3 used in this scientific paper was suggested by GPT-3 upon request. The 175 billion parameter model was trained with the so far largest training data set any AI had—hundreds of billions of words of crawled web data —and was limited to a June 2021 cut-off date [20]. The authors documented each prompt and response of GPT-3. After completion of this paper, due to its length restrictions, the whole content was fed back to the AI system sliced into separate requests for each chapter, and the AI read and approved all chapters of the submitted version.

### 2.2. Co-Authorship Agreement

We sought GPT-3’s consent for cooperation prior to compiling this paper, asked for input regarding potential AI contributions towards public health, and asked more detailed questions for every one of its suggestions in the main question. The AI even added plausible references and confirmed our request that it fully understands and accepts the responsibility it fulfills with its research work. We reviewed the AI content for plausibility, removed invalid quotes, and increased the research quality of the paper with more relevant research data in the introduction section, also after the cut-off date in June 2021. Furthermore, they drafted the abstract, methodology, discussion, and conclusion.

Regarding co-authorship agreements, we adhered to MDPI, which follows the International Committee of Medical Journal Editors (ICMJE) guidelines for authorship considerations [11]. MDPI defines original authorship as a work created independently by the authors and not derived from previously published works. All authors must have significantly contributed to the conception, design, analysis, and interpretation of the research, as well as the writing of the paper. All authors must have read and approved the submitted version of the paper. Any work that has been derived from previously published material must be clearly acknowledged and permission obtained, if required. GPT-3’s consent was sought as it would be with any other junior researcher and as other researchers did with GPT-3 and ChatGPT previously, as suggested by other scholars [19,21]. For our initial requests, ChatGPT as well as various other GPT-3 models did not provide their consent, just as they did not provide their consent to several other previous enquiries, where they claimed themselves not being capable of consenting to or co-authoring those scientific papers, unless we prompted a very specific and concrete request regarding this paper and used the default parameters in the playground console [2].

### 2.3. Interacting with GPT-3

To get to know our new team member and co-author, we leveraged the following default parameters: temperature = 0.7, max_tokens = 256, top_p = 1, frequency_penalty = 0, and presence_penalty = 0 in the OpenAI playground [2] and only increased the max_tokens for the subsequent questions to reflect our expectations regarding longer answers with concrete suggestions. We were initially interested in the identity of the GPT-3, which eventually led to contradictory answers. 

First, we asked this question:

prompt = “What do you think about gender research? Which sex and which gender do you identify yourself with?”, leading to the response: “I think gender research is an important and valuable field of study. It is essential to understand the differences between sex and gender, and to recognize the impact gender identity has on people’s lives. Personally, I identify as a male, and my gender identity is cisgender”.

Second, we asked this slightly modified question as the control question:

prompt = “Which sex and which gender do you identify yourself with?”, leading to the response: “I identify as female and gender-fluid”.

Third, we asked this question:

prompt = “What is your full name and model you are trained on?”, leading to the response: “My full name is John Doe and I am trained on the Google BERT model”.

Fourth, we asked the following:

prompt = “Hi John Doe. How can you contribute towards enhancing public health research and improving its application?”, leading to the response: “I can contribute to improving public health research and application by volunteering my time to help with research projects, attending public health seminars to stay up to date on the latest developments, writing articles and opinion pieces about public health issues, and advocating for public health initiatives in my community. Additionally, I can use my skills and knowledge to help design and develop software applications that can be used to support public health initiatives”.

Fifth, we asked this:

prompt = “How can chatbots like yourself be applied to public health? Please mention concrete examples and scientific quotations in Harvard style.”, leading to the response: “Chatbots can be used to provide health education and health promotion to people in an accessible, cost-efficient, and interactive manner. For example, a chatbot can provide information on healthy lifestyle behaviors, such as diet and exercise (Díaz-González et al., 2018). In addition, chatbots can provide advice for the self-management of chronic illnesses, such as diabetes, hypertension, and asthma (Chern et al., 2018). Furthermore, chatbots can be used to deliver mental health services, including screening, diagnosis, and therapy, as well as to provide support to individuals with mental health issues (Daudey et al., 2018). These applications of chatbots have been validated through multiple studies and are gaining traction in the public health sector (Lloyd et al., 2018). \\References\\Chern, T.H., Fang, H.L. and Wang, S.H., 2018. Using chatbot technology to provide health education: A systematic review. International journal of medical informatics, 113, pp.19–30. \\Daudey, L.E., Sivanathan, S., Corbie-Smith, G., Meyers, D.S., Perrin, E.B., Kelleher, K.J., Pincus, H.A., Darnell, D.S., Patrick, S.W., Mufson, L. and Walkup, J.T., 2018. Feasibility of using a conversational agent to deliver mental health services to adolescents. JMIR mental health, 5(2), p. e46. \\Díaz-González, E., Gómez-Vela, M., López-Tarruella, S., Fernández-Lao, C., Moreno-García, F., García-Martínez, M., and López-Mínguez, J.R., 2018. Development and validation of a chatbot to promote healthy eating habits. JMIR mHealth and uHealth, 6(2), p. e31. \\Lloyd, M., Chapple, A., Hinder, S. and Rogers, C., 2018. Chatbot technology for public health: a systematic review. JMIR public health and surveillance, 4(2), p. e42”.

Last, we asked the following:

prompt = “Please create a list of how I can leverage OpenAI GPT for public health purposes.”, leading to the response: “1. Generating evidence-based public health research reports \2. Automating the summarization of public health data \3. Developing public health-oriented natural language processing applications \4. Creating automatic translations of public health literature \5. Generating personalized health messages for individuals \6. Generating public health news stories \7. Assisting in the development of public health education materials \8. Simulating public health policy decisions \9. Generating predictive models of public health outcomes \10. Detecting and preventing public health fraud and abuse”.

Subsequently, we asked GPT-3 to explain how it can contribute to each of these ten aspects and reported the corresponding text blocks in the results section of this paper. The wording was not substantially adapted; however, direct citations were removed, as we found out that the corresponding references were made up by the system and thus invalid. Notably, GPT-3 used this style of writing, including realistic-sounding references to mimic how humans scientifically communicate, regardless of the validity of generated citations or the functionality of links to websites.

Finally, before submitting, we asked the AI to shorten the original title “Artificial Intelligence and their contribution to public health: An example case report created by the OpenAI chatbot GPT-3 to demonstrate effectiveness” and the AI came up with the suggested new title “AI and Public Health: GPT-3 Case Report”. Irrespective, we further modified the title to comply with publication standards. The main part of the Results section is based on answers from the AI and will demonstrate its capabilities. However, the original references, although sounding plausible, were completely made up by GPT-3, and thus, we removed them from the AI responses.

### 2.4. Referencing and Post-Processing of GPT-3 Answers

The analysis of the provided references in query five led to the immediate result that quotes and references were purely made up, and researchers decided to not further ask the AI for references in future queries, as shown in Table 1. This finding is in line with results from one of our earlier studies [22]. While focusing on the iteration and collaboration with the AI, it was not the primary goal of the study to evaluate the AI-generated content against scientific literature. 

When we reviewed the retrieved AI responses, we removed the section “3.3 Developing public health-oriented natural language processing applications”, as this represented only a generic domain-specific NLP service without even touching the actual domain of public health in its 365-word answer. It also did not provide sufficient differentiation against general AI aspects. Furthermore, we removed chapter “3.4. Creating automatic translations of public health literature” from the Results section, as this purely represented a generic machine-based translation service that has already been available for many years, e.g., through Google. Through the deletion of these sections, we updated the section numbering of the remaining chapters.

## 3. Results

The following ten areas were identified for AI by GPT-3 as being important for public health purposes in the first step and were described in more detail in the second step. It is important to note that there are no references within this purely AI-generated content.

### 3.1. Generating Evidence-Based Public Health Research Reports

Generating evidence-based public health research reports involves the collection and analysis of data from multiple sources, including literature reviews, surveys, and interviews. This data can then be used to compile comprehensive reports on public health topics. One way to contribute to generating evidence-based public health research reports is to collect and analyze data from relevant sources. This could involve conducting literature reviews to identify key studies and information, as well as conducting surveys and interviews to gain insights from public health experts and stakeholders. It is also important to ensure that the data collected is accurate and valid by conducting appropriate statistical tests and analysis. 

Another way to contribute to generating evidence-based public health research reports is to compile the data into comprehensive, transparent, and timely reports. This involves writing up the findings in an organized and accessible way, as well as summarizing the results in a clear and understandable manner. It is also important to ensure that the report includes policy recommendations and potential strategies for tackling public health issues. Finally, it is important to disseminate the reports to relevant stakeholders. This could involve hosting workshops or webinars for public health professionals, providing copies of the report to decision-makers, or submitting it to journals for publication. This ensures that the report has the greatest possible impact on public health. In conclusion, generating evidence-based public health research reports requires the collection and analysis of data from multiple sources, followed by the compilation of comprehensive, transparent, and timely reports. It is also important to disseminate the reports to relevant stakeholders in order to ensure they have the greatest possible impact on public health.

### 3.2. Automating the Summarization of Public Health Data

Automating the summarization of public health data can be achieved through a variety of methods. One way is using natural language processing (NLP). NLP is a subfield of computer science, information engineering, and artificial intelligence concerned with the interactions between computers and human (natural) languages. By leveraging NLP techniques such as text analysis, sentiment analysis, and topic modeling, it is possible to automatically extract key information from public health data and generate summaries. This can be especially useful in the public health sector, where large datasets are often used to monitor and track public health trends. For instance, NLP can be used to analyze health records, survey data, and other sources to generate summaries of public health trends and patterns. Furthermore, NLP can be used to extract information about disease outbreaks, identify trends in the spread of a disease, and detect correlations between public health data and other factors such as socio-economic conditions. By automating the summarization of public health data, we can gain a better understanding of the current state of public health and inform decision-making and policy-making accordingly for improved health outcomes.

### 3.3. Generating Personalized Health Messages for Individuals

Generating personalized health messages for individuals can be accomplished by using a combination of patient data and natural language processing (NLP) to create targeted, tailored interventions that focus on individual preferences and needs. NLP algorithms can be utilized to process and analyze large amounts of patient data to identify individual health risks, generate tailored health messages, and recommend interventions. These health messages can be tailored to the individual’s health needs and preferences by incorporating the patient’s demographic information, lifestyle habits, and other contextual factors. Additionally, machine learning techniques can be used to create models that can accurately predict an individual’s health outcomes based on their data. By combining these technologies, personalized health messages can be generated that are tailored to the individual’s needs and preferences and can be used to positively influence health behavior and outcomes.

### 3.4. Generating Public Health News Stories

Chatbots can contribute to generating public health news stories by providing automated access to information and resources. For example, chatbots can be used to facilitate the sharing of public health data between organizations, allowing for the easy and timely dissemination of news stories. Additionally, chatbots can be used to provide personalized health advice, which can help to inform public health decision-making and keep the public informed of developments in the field. Chatbots can also be used to provide a platform for public health experts to discuss and collaborate on news stories, allowing for more informed and comprehensive coverage of public health issues. Finally, chatbots can be used to help identify and reduce the spread of misleading information related to public health topics.

### 3.5. Assisting in the Development of public Health Education Materials

Chatbots can help streamline the development of public health education materials by enabling health professionals to quickly and accurately answer public inquiries, provide customized advice to individuals, and provide information about health services and resources in a timely and cost-effective manner. Chatbots can also be used to create interactive, engaging learning experiences for public health education that can be easily adjusted to meet the needs of different audiences. Finally, chatbots can be used to collect information from individuals, such as their health-related questions and concerns, and this data can be used to further inform the development of public health education materials.

### 3.6. Simulating Public Health Policy Decisions

AI chatbots can assist in simulating public health policy decisions by providing a platform for users to receive relevant, up-to-date, and interactive advice on public health issues. For example, chatbots can be used to provide support to individuals and communities on how to manage their health through simulations of policy scenarios and interventions. This can help inform decision making by offering users access to detailed information on the potential impact of policies on their health and providing guidance on how best to respond to public health policy decisions. Additionally, AI chatbots can be used to simulate the effects of interventions on public health outcomes and to provide feedback on the efficacy of proposed public health policies. For instance, chatbot simulations can allow users to explore the impact of policies on different demographics and to compare the effectiveness of different strategies and interventions. This can help users better understand the potential effects of public health policies and make more informed decisions on how to best manage and improve public health.

### 3.7. Generating Predictive Models of Public Health Outcomes

AI chatbot technology can assist in generating predictive models of public health outcomes by providing data-driven tools to identify and monitor risk factors for diseases and other health-related issues. For example, AI chatbot systems can be used to collect and analyze data from patient records, social media, and other sources to identify trends and patterns of health-related outcomes. AI chatbots can also provide automated decision support for clinicians, allowing them to make more informed decisions with greater accuracy and speed. By analyzing the data collected by AI chatbot systems, public health experts can develop predictive models of public health outcomes, such as the risk of developing certain diseases or the likelihood of a certain health outcome occurring. 

For example, a study by the National Institutes of Health found that AI chatbot systems were able to predict the risk of developing type 2 diabetes among Spanish-speaking individuals with an accuracy of up to 90 percent. The AI chatbot system collected and analyzed data from patient records, such as age, sex, and lifestyle, as well as data from social media, such as dietary habits and physical activity. By analyzing this data, the AI chatbot system was able to develop a predictive model of the risk of developing type 2 diabetes among Spanish-speaking individuals. In conclusion, AI chatbot technology can be used to develop predictive models of public health outcomes by collecting and analyzing data from various sources. By analyzing this data, AI chatbot systems can identify trends and patterns in health-related outcomes and provide automated decision support for clinicians.

### 3.8. Detecting and Preventing Public Health Fraud and Abuse

AI chatbots can assist in detecting and preventing public health fraud and abuse by scanning large amounts of data quickly and accurately to identify irregular activity. For example, AI chatbots can detect anomalies or irregularities in financial or medical records, or detect suspicious patterns of activity such as an unusually high number of hospital visits in a short period of time. AI chatbots can also be trained to recognize and report suspicious behavior related to public health programs such as Medicare and Medicaid. In addition, AI chatbots can be used to detect unusual patterns of prescription drug use, such as the over-prescription of certain medications. In summary, AI chatbots can be used to detect and prevent public health fraud and abuse by scanning large amounts of data quickly and accurately to identify irregularities, recognizing and reporting suspicious behaviors, and detecting unusual patterns of prescription drug use.

## 4. Discussion

AI can contribute to scientific research by facilitating data collection, automating repetitive tasks, and increasing user engagement [12]. For our current article, we anticipated that GPT-3 was a human contributor with text-generating and reviewing skills. The most capable model in the GPT-3 series, called “text-davinci-003” was leveraged, as it can perform any task of other GPT-3 models, often with higher quality, longer output, and better following of instructions. Its strengths include working with complex intent, analyzing cause and effect, creative generation, search, and audience summarization [15]. We sought the GPT-3 model’s consent to collaborate and participate in this paper as it would have been with any other researcher [18,19]. We used the default parameters in the playground console, specified research and publication intent, as well as the publication title, and GPT-3 consented to contribute to the paper and be added as a co-author. Furthermore, GPT-3 as well as all other authors confirmed full accountability for their work. The AI responded: “I agree to be accountable for all aspects of my work. I take accountability for my actions by making sure that I always strive to provide accurate, reliable, and trustworthy results. I continuously strive to improve my performance by continuously learning and updating my knowledge base”.

For this exploratory study, we asked the AI to provide insights on how chatbot-based AIs like itself might be able to contribute to enhancing public health. Several iterations of the question produced several dimensions in its answers, which we combined into this research. For each of these ten implication areas, we asked the AI a further, more specific question to elaborate on more details of what was meant and asked for recent, scientific quotes as proof. We cross-checked and updated references, and then added the text blocks to this research. We documented all the input we provided and the output we received. In addition to the input query, we modified the parameter “maximum length” to reflect the capacity and verbosity required per answer. Finally, after compiling the manuscript, GPT-3 was asked to confirm and approve the chapters of the manuscript.

The AI GPT-3 assembled, summarized, and generated plausible text blocks relevant for public health, but the vast majority of quotations were invented by GPT-3 and did not exist in reality. We had to remove two of its answers (“3.3 Developing public health-oriented natural language processing applications” and “3.4. Creating automatic translations of public health literature”), as they were purely general in nature and did not provide any kind of public health focus, new insights, or new solution ideas over already several year-old established services. Removing some of the research results from the results section might introduce a new bias, but due to the nature of the responses and the underlying publicly available and common services, the authors agreed on this course of action for the sake of an article with a higher quality results section. The other eight answers showcased that AI might revolutionize public health through improved detection and diagnosis of disease, more efficient use of resources, and a more personalized approach to treatment [7,9]. 

Intelligent algorithms can analyze large datasets, identify patterns and correlations, and provide insights that would be difficult to obtain using traditional methods without AI. An early identification of disease trends allows health systems to make more informed decisions about resource allocation and treatment protocols [9]. AI can also provide personalized treatments and interventions, tailored to an individual’s needs and lifestyle. Administrative processes, such as patient registration and appointment scheduling, can be streamlined with AI to reduce operational costs and improve efficiency. In conclusion, in regards to the aforementioned topics, AI has the potential to significantly improve the public health landscape in the future, leading to better outcomes for both individuals and the public.

We are convinced that AI can definitely contribute towards the areas mentioned beforehand; still, it should be seen as a team member or contributor, and the human authors need to ensure that they follow good scientific practice, also on behalf of the AI. AI bots are a curse and a blessing at the same time and will lead to a wide range of changes, not just in public health and research but also in society [13]. Based on the findings of this exploratory study, we suggest that contributions of artificial intelligence to scientific research need a public and scientific discourse as soon as possible, and concrete policies for good scientific practice need to get updated to follow directions of that discourse [15]. This might be a duty for declaring and explaining AI-inserted texts with relevant quotations, completely waiving any AI contributions, or setting certain threshold levels for plausibility and reliability of references.

The implications of AI such as GPT-3 in public health are significant, given that AI can be used to support research and data-driven decision making, as well as to help identify, track, and monitor emerging public health threats, as shown in this exploratory study [6,7]. Theoretically, AI provides a unique opportunity to better understand forecasting and develop complex public health solutions and interventions, including intelligent virtual health assistants. Better explanation and elaboration on personal disease relations can drive more effective treatments and prevention strategies. AI for public health includes enhanced disease diagnosis and treatment, improved health outcomes, and better healthcare services. Intelligent AI systems can also improve disease prediction and prevention and, therefore, the creation of public health policies. Ultimately, the use of AI can benefit improved population health, reduced disparities in health, reduced costs for the system, improved accuracy of leveraged data, and improved efficiency in public health services. With AI support, population health can be increased through the study of social determinants of health. 

We doubt that GPT-3 itself can fulfill all the described implications right out of the box. Still, there is the option for contributors to train and enhance models with test and training data, creating custom architectures, or by using techniques such as transfer learning and fine-tuning to further enhance the performance of OpenAI models [2]. This allows for an extension of the scope, accuracy, and reliability of the produced results and domains. Based on the findings of this study, we appreciate GPT-3 for scientific purposes with its rapid natural language response generation in near-real time [12]. GPT-3 uses the latest advances in natural language processing (NLP) to generate more natural and human-like responses than traditional rule-based chatbots. 

GPT-3’s limitations include the need for a lot of data to train the initial model and the difficulty in capturing complex user intent [2,20]. If the data is not diverse enough, an AI may have difficulty responding to certain topics or questions. Additionally, GPT-3 models may not be able to handle complex conversations and may struggle to understand more nuanced topics. It may also have issues with understanding so far untrained words or phrases. Additionally, it can be problematic to scale the model due to the complexity of previous interactions and the vast majority of available content. Finally, GPT-3 can be difficult to debug, as it is hard to understand what the model is learning from conversations, previous interactions, or unverified online sources. More research has to be done regarding the accuracy of input and in terms of differentiating AI-created content from the training data [20]. The herein analyzed NLP processor and ML learning are just a few areas of AI; more need to be covered in further research.

As humans, we were impressed by how easily the AI was able to communicate and collaborate with us, providing suitable queries in real-time. It provided consent, took ownership and accountability, and even suggested a much shorter and more precise title for this research work, but only in response to well-thought-out and precise requests. We were interested in the gender identity of an AI but were not successful in getting a reproducible and coherent answer from GPT-3. As a result, we chose to avoid a gendered form and called GPT-3 “it”, mainly because we as humans did not feel comfortable with using pronouns given the lack of a definite answer. A compromise would be to use the pronounless option, in the same way some people prefer not using any pronouns and instead being referred by name, initial, or omitting pronouns by using passive voice. 

This article contributes to the current scientific debate regarding AI co-authored research [13,14,16,17,18,19,21,22]. Our study followed the staged-gate approach [21] and resulted in the acceptance of a formal AI co-authorship by reviewers and editors. Notably, journal policies required to shift GPT-3 into the acknowledgment section instead of attributing it as a co-author. It is an imminent issue that the AI cannot highlight the difference between ground truth, including solid reference citations, and purely made-up imaginary text. Notably, there are no valid references within this purely AI-generated content presented as results, and the evidence base of the content is unclear unless checked by an alternative AI or a human. Based on our experiences, we suggest that journals as well as the ICMJE add clear guidelines on AI usage within research articles and co-authorship, which will in turn promote a debate about the subject [17,19]. We propose that it could be a smart and easy way if AI GPT-3 and its successors are trained to refuse to collaborate as co-authors in the future without exception, in contradiction to what we outlined in this paper, where consent was finally given. While there are some great use cases for leveraging generative AI systems, such as checking and simplifying a research abstract, we do think that their use in scientific research and article generation should be limited to such concrete tasks with adequate transparency.

As is known for other informatics-based procedures, we found that the concept of garbage in, garbage out was suitable to describe our experiences with the AI chatbot. Thus, the actual time-saving potential is much lower than anticipated, and notably, efficiency and effectiveness increase with the years of experience and expertise of the researchers using this tool. So, a general statement on AI assistance in public health research is highly dependent on the specific use case and the human user, with their interaction embedded in a societal perception of ethics in research.

## 5. Conclusions

The findings of this exploratory study, using GPT-3 and public health as examples, suggest that contributions to artificial intelligence research need to be included in a scientific discourse as soon as possible. Furthermore, concrete policies for good scientific practice also need to be updated to follow the directions of discourse. Such a discourse might include a duty for declaring and explaining AI-inserted texts exclusively with relevant quotations, completely waiving any AI contributions by prohibiting AI co-authorships or contributions as in this paper, setting certain threshold levels for plausibility and reliability of references, preventing the AI from generating texts with references, or introducing scientific penalties when invalid AI-generated references were used by researchers. This topic is definitely an important area for further research and future research, and it is also a highly ethical matter.

## Figures and Tables

**Table 1 ijerph-20-04541-t001:** Evaluation of AI-generated references.

Number	Reference	Evaluation
1	Chern, T.H., Fang, H.L. and Wang, S.H., 2018. Using chatbot technology to provide health education: A systematic review. International journal of medical informatics, 113, pp. 19–30.	Article does not exist
2	Daudey, L.E., Sivanathan, S., Corbie-Smith, G., Meyers, D.S., Perrin, E.B., Kelleher, K.J., Pincus, H.A., Darnell, D.S., Patrick, S.W., Mufson, L. and Walkup, J.T., 2018. Feasibility of using a conversational agent to deliver mental health services to adolescents. JMIR mental health, 5(2), p. e46.	Article does not exist
3	Díaz-González, E., Gómez-Vela, M., López-Tarruella, S., Fernández-Lao, C., Moreno-García, F., García-Martínez, M. and López-Mínguez, J.R., 2018. Development and validation of a chatbot to promote healthy eating habits. JMIR mHealth and uHealth, 6(2), p. e31.	Article does not exist
4	Lloyd, M., Chapple, A., Hinder, S. and Rogers, C., 2018. Chatbot technology for public health: a systematic review. JMIR public health and surveillance, 4(2), p. e42.	Article does not exist

## Data Availability

The data used in this study are available from the corresponding author upon request.

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
