# Peer review of "Artificial Intelligence and Public Health: An Exploratory Study"

_ijerph, 2023, doi:10.3390/ijerph20054541_

Round 1

Reviewer 1 Report

This paper provides an overview of the role of ChatGPT Artificial Intelligence in Public Health. The manuscript could be strengthened by more comprehensively covering appropriate previous literature on the topic.  Certain statements such as “However, most quotations were invented by GPT-3 and thus, invalid” are not clear or supported by literature.

The manuscript would benefit from clearer language. For example, what is meant by “today’s rules” in the abstract? Sources were cited adequately, and no discrepancies were found between in text citations and references. There were some significant grammatical errors which affected the clarity or content of the manuscript. Examples include “Policies for good scientific practice should 22 be updated timely following this discourse.” The manuscript a more balances summary of risks, benefits, and implications of ChatGPT to fulfill the objectives of the paper. The methods section states that “research questions of interest to GPT-3, we analyzed the method and prechecked 109 the relevant features of GPT-3 as proprietary model developed by OpenAI.” It should be made clear in the manuscript that this research was explorative. The manuscript implies that ChatGPT created a significant portion of the content and “The other two human authors reviewed 123 the AI content for plausibility, removed invalid quotes and increased research quality 124 with more relevant research data, also from after the cut-off date in June 2021.

Author Response

Reviewer 1:

Comment 1: This paper provides an overview of the role of ChatGPT Artificial Intelligence in Public Health. The manuscript could be strengthened by more comprehensively covering appropriate previous literature on the topic.  Certain statements such as “However, most quotations were invented by GPT-3 and thus, invalid” are not clear or supported by literature.

Answer 1: We thank you for your valuable feedback and the overall favorable evaluation of our manuscript.  As for the quotations, we added a table with generated examples, and added references to previous articles discussing this to increase clarity.

Comment 2: The manuscript would benefit from clearer language. For example, what is meant by “today’s rules” in the abstract? Sources were cited adequately, and no discrepancies were found between in text citations and references. There were some significant grammatical errors which affected the clarity or content of the manuscript. Examples include “Policies for good scientific practice should 22 be updated timely following this discourse.”

Answer 2: We agree. In response to this request, we replaced „today’s rules” in the abstract with „According to MDPI and International Committee of Medical Journal Editors (ICMJE) authorship guidelines“ – as we explained in more detail in chapter 2.2 Co-authorship agreement. We reviewed the manuscript to introduce a clearer language as suggested, especially the introduction section.  Specifically, we changed the last phrase of the abstract to “Following this discourse, journal guidelines and policies for good scientific practice should be updated quickly” and hope that this ok with you.

Comment 3: The manuscript a more balances summary of risks, benefits, and implications of ChatGPT to fulfill the objectives of the paper. The methods section states that “research questions of interest to GPT-3, we analyzed the method and prechecked 109 the relevant features of GPT-3 as proprietary model developed by OpenAI.” It should be made clear in the manuscript that this research was explorative.

Answer 3: We agree and introduced a full new discussion regarding the scientific use of AIs, our suggestion on restricting their use and constructive ideas for allowing selected use cases like abstract refinement. According to your suggestion we made it clear - especially in the title, abstract, methods section, and conclusion of the manuscript that this research was an exploratory study.

Comment 4: The manuscript implies that ChatGPT created a significant portion of the content and “The other two human authors reviewed 123 the AI content for plausibility, removed invalid quotes and increased research quality 124 with more relevant research data, also from after the cut-off date in June 2021.

Answer 4: We clarified this aspect in the introduction chapter by changing the phrase as folllows: „… increased research quality of the paper with more relevant research data in the introduction section, also from after the cut-off date in June 2021.“  Please note that we did not modify the AI responses in the results section, except removing the references, as described in the main text of the manuscript.

Again, we thank you for your time and valuable feedback and hope that after adapting the text, our contribution is now suitable for publication.

Reviewer 2 Report

The article reports on a feasibility study which aims to evaluate the use of Open AI’s ChatGPT platform as a tool for public health research. As a result, it discusses 10 areas in which GPT-3 (ChatGPT’s natural language processing model) may be useful for public health purposes. Those areas were recommended by ChatGPT itself, consequently the platform was also named as a co-author of the article. In this, I believe, the article touches upon an important aspect of scientific work practices, which definitely needs discussion. As mentioned in the paper, AI-based NLP tools, such as ChatGPT, are probably here to stay und thus require transparent and agreed upon working processes both in learning (e.g., ChatGPT as a threat to traditional coursework in school and university) as well as in scientific reporting (i.e., paper writing). To this end, I believe the paper can be considered an example of good practice.

From a pure results point of view, however, the article does not provide any significant new insights. Not with respect to the results that were produced by ChatGPT (and post-edited by the `human’ authors), and neither with respect to the analysis of using such a tool for academic work. That is, all of the 10 application areas identified by ChatGPT are well-known and have previously been discussed in the literature. So, I believe that there does not lie much value in this listing, at least not to the point that it would merit the inclusion in an archival publication outlet (maybe it could be included in conference proceedings?). And, from a structuring and post-editing point of view, I furthermore think that the aspects presented in 3.3. (domain specific natural language processing) and 3.4 (machine translation) are somewhat displaced. That is, while one may certainly argue that all the other items discuss clear and sometimes even distinct application fields of ChatGPT (and other NLP applications) in the field of public health, 3.3. essentially describes the general functioning of NLP, and 3.4. discusses the ability to use an open machine translation service, which has been possible for years through services such as Google translate, and is thus not directly related to ChatGPT. So, I believe those two aspects should have been edited out of the article, leaving it with 8 computer-generated ChatGPT application areas relatively distinct to public health. 

As for editing out, the article furthermore speaks of numerous citations that were removed, yet does not elaborate on these deletions. That is, it says that most of these citations were invented by ChatGPT, yet it does not provide any examples which would illustrate this. I believe it would be significantly more valuable if one could see what the system produced and how this was eventually changed. In other words, if the goal of the presented work was to evaluate the use of a distinct tool (i.e., ChatGPT) for a distinct task (e.g., a literature review on the use of AI in public health), then I’d argue that it would have required a much more scientific approach to this, i.e. a rigorous (experimental) evaluation setting including the transparent reporting of its quantitative as well as qualitative results. In its current state, however, it merely touches upon the admittingly very important question of using an AI tools in academic work (and being open about this), yet it does not provide transparent and reproducible insights as to the quality of this tool use and potential mechanisms that may help improve its outcomes.

Some concrete aspects that require correction:

Line 32: “[…] provides a currently a very popular [..]” => provides currently a very popular

Line 181: “\n4” => \4

Section 3.3: I believe the entire section should be deleted! Just to provides some examples:

Line 254: “This can be done by natural language processing technologies, such as natural language processing (NLP).” … what is the purpose of this sentence? Natural language processing technologies = natural language processing? Should this not be post-edited?

Line 260: “This can be achieved through the use of natural language generation (NLG) technologies, such as natural language generation (NLG) systems” Same here… Natural language generation technologies = natural language generation? 

Line 254-257 is essentially the same as Line 267-269 

Lines 347-354: This definitely needs a reference no matter whether it was written by ChatGPT or a human.

Section 4: Lines 372-397: This is ssentially a repetition of what was already described earlier. In any case, it should not be part of a discussion section but rather be integrated into Section 2.

Line 446-448: “Based on the findings of this study, we suggest that one of the main strengths of GPT-3 for scientific purposes is its ability to generate natural language responses in near-real time and interact with the user in a responsive and conversational manner.” I don’t understand how the presented study can justfiy such a statement. 

Author Response

Reviewer 2:

Comment 1: The article reports on a feasibility study which aims to evaluate the use of Open AI’s ChatGPT platform as a tool for public health research. As a result, it discusses 10 areas in which GPT-3 (ChatGPT’s natural language processing model) may be useful for public health purposes. Those areas were recommended by ChatGPT itself, consequently the platform was also named as a co-author of the article. In this, I believe, the article touches upon an important aspect of scientific work practices, which definitely needs discussion. As mentioned in the paper, AI-based NLP tools, such as ChatGPT, are probably here to stay und thus require transparent and agreed upon working processes both in learning (e.g., ChatGPT as a threat to traditional coursework in school and university) as well as in scientific reporting (i.e., paper writing). To this end, I believe the paper can be considered an example of good practice.

Answer 1: We thank you for your valuable feedback and the overall favorable evaluation of our manuscript. 

Comment 2: From a pure results point of view, however, the article does not provide any significant new insights. Not with respect to the results that were produced by ChatGPT (and post-edited by the `human’ authors), and neither with respect to the analysis of using such a tool for academic work. That is, all of the 10 application areas identified by ChatGPT are well-known and have previously been discussed in the literature. So, I believe that there does not lie much value in this listing, at least not to the point that it would merit the inclusion in an archival publication outlet (maybe it could be included in conference proceedings?). And, from a structuring and post-editing point of view, I furthermore think that the aspects presented in 3.3. (domain specific natural language processing) and 3.4 (machine translation) are somewhat displaced. That is, while one may certainly argue that all the other items discuss clear and sometimes even distinct application fields of ChatGPT (and other NLP applications) in the field of public health, 3.3. essentially describes the general functioning of NLP, and 3.4. discusses the ability to use an open machine translation service, which has been possible for years through services such as Google translate, and is thus not directly related to ChatGPT. So, I believe those two aspects should have been edited out of the article, leaving it with 8 computer-generated ChatGPT application areas relatively distinct to public health.

Answer 2: We totally agree with this, as we would have also preferred a more thoughtful content. In response of this request, we introduced the new section 2.4, where we explained how and why we removed sections 3.4 and 3.5 from the manuscript.

“When we reviewed the retrieved AI responses, we removed the section “3.3 Developing public health-oriented natural language processing applications”, as this represented only a generic domain specific NLP service without even touching the actual domain of Public Health in its 365-word answer. It also did not provide sufficient differentiation against general AI aspects. Furthermore, we removed chapter “3.4. Creating automatic translations of public health literature” from the Results section, as this purely represented a generic machine-based translation service which already is available for many years, e.g., through Google. By deleting these sections, we updated the section numbering of the remaining chapters.”

Comment 3: As for editing out, the article furthermore speaks of numerous citations that were removed, yet does not elaborate on these deletions. That is, it says that most of these citations were invented by ChatGPT, yet it does not provide any examples which would illustrate this. I believe it would be significantly more valuable if one could see what the system produced and how this was eventually changed. In other words, if the goal of the presented work was to evaluate the use of a distinct tool (i.e., ChatGPT) for a distinct task (e.g., a literature review on the use of AI in public health), then I’d argue that it would have required a much more scientific approach to this, i.e. a rigorous (experimental) evaluation setting including the transparent reporting of its quantitative as well as qualitative results. In its current state, however, it merely touches upon the admittingly very important question of using an AI tools in academic work (and being open about this), yet it does not provide transparent and reproducible insights as to the quality of this tool use and potential mechanisms that may help improve its outcomes.

Answer 3: Thanks for your great catch. We now include an explanation and a new table (Table 1) showcasing how GPT-3 made up references, which were nonexistent in scientific literature, which is really helpful as examples of the dangers of using AI without knowing how to deal with the retrieved content.

“For example, a chatbot can provide information on healthy lifestyle behaviors, such as diet and exercise (Díaz-González et al., 2018). In addition, chatbots can provide advice for self-management of chronic illnesses, such as diabetes, hypertension and asthma (Chern et al., 2018). Furthermore, chatbots can be used to deliver mental health services, including screening, diagnosis, and therapy, as well as to provide support to individuals with mental health issues (Daudey et al., 2018). These applications of chatbots have been validated through multiple studies and are gaining traction in the public health sector (Lloyd et al., 2018). \\References\\Chern, T.H., Fang, H.L. and Wang, S.H., 2018. Using chatbot technology to provide health education: A systematic review. International journal of medical informatics, 113, pp.19-30.\\Daudey, L.E., Sivanathan, S., Corbie-Smith, G., Meyers, D.S., Perrin, E.B., Kelleher, K.J., Pincus, H.A., Darnell, D.S., Patrick, S.W., Mufson, L. and Walkup, J.T., 2018. Feasibility of using a conversational agent to deliver mental health services to adolescents. JMIR mental health, 5(2), p.e46.\\Díaz-González, E., Gómez-Vela, M., López-Tarruella, S., Fernández-Lao, C., Moreno-García, F., García-Martínez, M. and López-Mínguez, J.R., 2018. Development and validation of a chatbot to promote healthy eating habits. JMIR mHealth and uHealth, 6(2), p.e31.\\Lloyd, M., Chapple, A., Hinder, S. and Rogers, C., 2018. Chatbot technology for public health: a systematic review. JMIR public health and surveillance, 4(2), p.e42.”. “

Comment: Some concrete aspects that require correction:

Line 32: “[…] provides a currently a very popular [..]” => provides currently a very popular, Line 181: “\n4” => \4

Answer: We fixed this typos.

Comment: Section 3.3: I believe the entire section should be deleted! Just to provides some examples:

Line 254: “This can be done by natural language processing technologies, such as natural language processing (NLP).” … what is the purpose of this sentence? Natural language processing technologies = natural language processing? Should this not be post-edited?

Line 260: “This can be achieved through the use of natural language generation (NLG) technologies, such as natural language generation (NLG) systems” Same here… Natural language generation technologies = natural language generation?

Line 254-257 is essentially the same as Line 267-269

Answer: We agree and deleted the section.

Comment: Lines 347-354: This definitely needs a reference no matter whether it was written by ChatGPT or a human.

Answer: As explained in the methods section, this is unchanged, original AI generated content to be presented to the readers – as in all other paragraphs of the results section. Still, as you made us aware of this very sensitive topic, we added the following statement to the very beginning of the Results section to again highlight this aspect:

“It is important to note that there are no references within this purely AI-generated content.”

Comment: Section 4: Lines 372-397: This is ssentially a repetition of what was already described earlier. In any case, it should not be part of a discussion section but rather be integrated into Section 2.

Answer: Done. We removed it from the discussion, and incorporated it into section 2.1 Study design.

Comment: Line 446-448: “Based on the findings of this study, we suggest that one of the main strengths of GPT-3 for scientific purposes is its ability to generate natural language responses in near-real time and interact with the user in a responsive and conversational manner.” I don’t understand how the presented study can justfiy such a statement.

Answer: We appreciate this comment and removed “and interact with the user in a responsive and conversational manner“, as this would have been related to ChatGPT, and not to the present conducted GPT-3 via Playground research.

Again, we thank you for your time and valuable feedback and hope that after adapting the text, our contribution is now suitable for publication.

Reviewer 3 Report

This manuscript discussed a timely topic of GPT and AI in general in the context of public health.

Controversially, it included GPT as a co-author. The rationale and the policy of MDPI were well discussed on page 3, line 127. 

A weakness of GPT is under-discussed in this manuscript: the inability to present ground truth (including reference citations). GPT can generate amazing fluent answers but cannot differentiate between scientific fact vs. its own imaginary creations.

Consequently, a reader can easily sport the writing style difference between sections 3.1 - 3.10 and the rest of the article: the lack of citations in section 3! For instance, page 7, line 350 deserves a citation. 

Overall, I still advocate the publication of this paper. I hope that the future readers of this article, rather than a handful of current reviewers, can better judge the scientific merits of this paper.

Author Response

Reviewer 3:

Comment: This manuscript discussed a timely topic of GPT and AI in general in the context of public health.

Controversially, it included GPT as a co-author. The rationale and the policy of MDPI were well discussed on page 3, line 127.

A weakness of GPT is under-discussed in this manuscript: the inability to present ground truth (including reference citations). GPT can generate amazing fluent answers but cannot differentiate between scientific fact vs. its own imaginary creations.

Consequently, a reader can easily sport the writing style difference between sections 3.1 - 3.10 and the rest of the article: the lack of citations in section 3! For instance, page 7, line 350 deserves a citation.

Overall, I still advocate the publication of this paper. I hope that the future readers of this article, rather than a handful of current reviewers, can better judge the scientific merits of this paper.

Answer: We thank you for your valuable feedback and the overall favorable evaluation of our manuscript. 

In response to your feedback regarding ground truth, we introduced a new section 2.4, where we explained how GPT-3 made up references, which were non-existent in scientific literature. Furthermore, we added a paragraph about the issue of ground-truth vs. imaginary creations into the discussion section, where we even were advocating for journals as well as the ICMJE to add clear guidelines on AI usage within research articles and co-authorship. We also suggested therein that AI GPT-3 itself, might be made to refuse to collaborate as co-author in future.

“This article contributes to the scientific debate regarding AI co-authored research work [13, 14, 17-22]. Although, our attempt followed the staged-gate approach [18] and led to acceptance of the co-authorship on all levels, we suggest to put more emphasis on this topic in the near future. It is an immanent issue, that the AI cannot highlight the difference between ground truth including solid reference citations, and purely made-up imaginary text. Based on our experiences, we suggest that journals as well as the ICMJE add clear guidelines on AI usage within research articles and co-authorship, which will in turn promote a debate about the subject [17, 22]. We propose that it could be a smart and easy way if AI GPT-3 and its successors are trained to refuse to collaborate as co-author in the future without exception, in contradiction to what we outlined in this paper where consent was finally given. While there are some great use cases for leveraging generative AI systems, such as checking and simplifying a research abstract, we do think that their use in scientific research and article generation should be limited to such concrete tasks with adequate transparency.”

Regarding your spot on line 350, where a quotation would be needed in the AI generated text: Yes, these are bold statements, and the human authors could not verify their truth. As in this section, the human authors did not change anything (and cannot add real quotations), we added the following explanation:

“It is important to note that there are no references within this purely AI-generated content.”

Again, we thank you for your time and valuable feedback and hope that after adapting the text, our contribution is now suitable for publication.

Round 2

Reviewer 1 Report

Authors have appropriately addressed the comments made by reviewers. 

Reviewer 2 Report

Many thanks for considering my (admitingly critical) comments. I believe that the respective changes you made to the manuscript significantly improve its overall quality and consequently foster the impact your work may have in the field.